# Microwave Planar Resonant Solutions for Glucose Concentration Sensing: A Systematic Review

Carlos G. Juan [1,*], Benjamin Potelon [1,*], Cédric Quendo [1] and Enrique Bronchalo [2]

1   Lab-STICC, Université de Bretagne Occidentale, 29200 Brest, France; cedric.quendo@univ-brest.fr
2   Department of Communications Engineering, Miguel Hernández University of Elche, 03202 Elche, Spain; ebronchalo@umh.es
*   Correspondence: carlosgabriel.juanpoveda@univ-brest.fr (C.G.J.); benjamin.potelon@univ-brest.fr (B.P.)

**Featured Application:** Updated review of microwave planar resonant sensors for glucose concentration tracking, sorted by sensing parameter and including relative sensitivity to glucose.

**Abstract:** The measurement of glucose concentration finds interesting potential applications in both industry and biomedical contexts. Among the proposed solutions, the use of microwave planar resonant sensors has led to remarkable scientific activity during the last years. These sensors rely on the changes in the dielectric properties of the medium due to variations in the glucose concentration. These devices show electrical responses dependent on the surrounding dielectric properties, and therefore the changes in their response can be related to variations in the glucose content. This work shows an up-to-date review of this sensing approach after more than one decade of research and development. The attempts involved are sorted by the sensing parameter, and the computation of a common relative sensitivity to glucose is proposed as general comparison tool. The manuscript also discusses the key points of each sensor category and the possible future lines and challenges of the sensing approach.

**Keywords:** glucose sensor; insertion/return loss; microwave; phase measurement; planar; relative sensitivity; resonant frequency; resonator; unloaded quality factor

## 1. Introduction

The use of microwave techniques for developing all sort of sensors has stirred an intense research field, which has been attracting increasing attention by the scientific community during the last decades. The efforts devoted to these techniques have generally paid off, leading to successful application to many contexts (e.g., [1–5]). Although many approaches can be taken, the common measurement principle of these sensors is the electromagnetic interaction between the sensor and the medium under test. The propagation of electromagnetic fields through the media depends, among other parameters, on the permittivity of the media, a frequency-dependent parameter unique for each material. Many microwave devices show responses strongly dependent on the permittivity of the surrounding media. This, in addition to their ease of integration and cost-effectiveness, explains why these devices are often used as permittivity sensors.

Several techniques can be used for this dielectric characterization purpose. Depending on the medium under test, application and frequency range, different methods are employed, such as wideband antennas [1], coaxial lines [6], planar capacitive techniques [7], frequency synthesizer-based methods [8] or CMOS microwave sensors [9], among others. Due to several interesting features, such as cost-effectiveness, ease of development and ease of integration, methods based on planar microwave resonators are frequently considered for applications requiring dielectric permittivity measurements [10–12], especially for

characterization of lossy organic liquids [13]. It is in this context where the attention of a part of the microwave community was attracted towards glucose sensing.

Glucose concentration sensing finds potential application in several contexts. The benefits for the diabetes community would be evident. People with diabetes need to self-monitor their blood glucose level several times every day (desirably with continuous monitoring) so that corrective actions can be accordingly applied. This technology could potentially lead to more comfortable and convenient measurement systems, notably helping to alleviate this burden [14]. Outside of the diabetes community, but still within a health context, glucose tracking is also of interest as a key indicator for other disorders [15,16], and a large enough surveillance could even be interesting for early detection and stopping of pandemic outbreaks [17]. In addition, glucose gauging also finds application in some industry processes [18], such as those dealing with the production of glucose-containing drinks (juices, sodas, beers, wines, spirits and so on). The current methods in these processes usually involve chemical probes [19,20], and therefore inline sensors able to provide a continuous measurement of the glucose content of the product would be of interest.

Several measuring technologies have been proposed for different contexts, such as electrochemical [21–24] or optoelectronic devices [25,26], among others. However, microwave devices, and especially planar resonant ones, show some beneficial features such as ease of production and integration, reduced size, competitive cost and interesting penetration depths, especially for non-invasive measurements, generally with low tissue scattering [27]. Under these circumstances, the study of microwave planar resonant techniques for glucose concentration sensing has become an intense research field during the last years. As a matter of fact, searching the keywords "microwave", "resonator" and "glucose" in the Scopus database yields 127 results at the moment of writing this article, 70.16% of them being from the last 5 years. For such a highly specific technical topic, these numbers give a clear idea of the current interest raised within the scientific community.

Indeed, the recent trends point to already attained interesting sensitivities (still to be enhanced, though), and pose the selectivity, i.e., the ability of the device to respond to changes only in the glucose concentration without interference from other elements, as one of the main current challenges [28,29]. Effectively, these devices are usually affected by undesired sensitivities to other physiological parameters [30–32]. While sensitivity can be enhanced by active circuitry techniques [33–36], selectivity seems only to be addressable by acquisition of redundant information by means of multi-sensor or multi-parameter measurements [28,31,37–39]. The advancements in machine-learning techniques cast growing optimism on these multi-parameter fusion approaches [40–43], which could lead to considerable gains of selectivity and robustness [38,44–46]. In this framework, looking at the literature, one can find up-to-date reviews of planar microwave sensors for general dielectric permittivity measurements [47–49], electromagnetic sensors for general biomedical [50,51] or industry [52] applications, and multitechnology sensors for glucose sensing in both health and industry contexts [28,29,53–58], but not a specialized review of planar microwave resonant glucose sensors. Considering the attention raised by this specific kind of sensor and the number of references available, the latter seems convenient.

This manuscript offers a systematic review of the available approaches to glucose sensing, regardless of the application scenario, with the specific focus on planar resonant methods. A new classification scheme based on the sensing parameter, suitably adapted for this sensing paradigm, is proposed, and the review is thereby organized accordingly. To ease the understanding and comparison of the different approaches involved, the computation of a normalized relative sensitivity to glucose is proposed, which is especially convenient for discussion involving different sensing parameters. The manuscript is organized as follows. Section 2 outlines the fundamentals of this sensing technique and its development from the early stages to the current attempts. It also discusses the categorization possibilities and the finally proposed classification criterion, as well as the calculation of a general sensitivity. Sections 3–6 review the different available works within each category, whereas Sections 7 and 8 show the discussion of the current state of this technology and draw the main conclusions of the study.

## 2. Microwave Planar Resonant Glucose Sensors: Fundamentals and Classification

*2.1. Approach to Microwave Planar Resonant Glucose Sensors*

The utility of microwave techniques for the measurement of glucose concentration is due to the influence of this magnitude in the complex permittivity of water, blood and other aqueous solutions [59]. The presence of glucose affects the orientational polarization process of water, shifting the relaxation spectrum towards lower frequencies [60–62]. The influence of glucose concentration in the permittivity of water–glucose solutions has been reported for a wide variety of frequency ranges using different techniques (e.g., [63–67]). This dielectric signature of glucose can find applications in the industrial production of sugar-containing drinks. In the diabetes community, the effect of glucose on the permittivity of blood [68,69] and plasma [70] opens the way for the design of non-invasive glucose sensors [71], although there remain serious technological problems to be solved in this field.

Within the common microwave techniques, planar devices have received the greatest attention for this concern. They are usually considered for many applications due to their cost-effectiveness and ease of integration. However, other approaches have also been studied, such as waveguide techniques [72,73], although their bulky nature hinders their evolution towards feasible commercial devices. Among the possibilities of planar technology for glucose sensing, resonant sensors are the most frequently found in the scientific literature, although other approaches have been studied, such as microstrip lines in transmission/reflection modes [74] or planar antennas in transmission [75,76] or reflection modes [77]. Resonant planar sensors present the advantage of concentrating the electric field in a small volume at frequencies around the resonant frequency, thus intensifying the sensor–MUT (Material Under Test) interaction. Additionally, they require a narrow bandwidth due to their sharp electrical response, which facilitates the design of the associated electronics. Since the first proposals 30 years ago [78], planar resonators have been widely used for dielectric permittivity measurements in a plethora of contexts.

As a simple example of resonant glucose sensor, let us consider a planar resonator loaded with a dielectric sample (an aqueous glucose solution). This can be represented by the parallel *RLC* lumped-element model shown in Figure 1, where the subscripts "r" and "s" make reference to the contribution of the resonator itself and the sample to each element, *J* elements are admittance inverters modeling the I/O couplings and $P_1$ and $P_2$ are the input and output ports, respectively. The response of a microwave resonator is generally defined by means of the resonant frequency ($f_r$) and unloaded quality factor ($Q_u$). Applying classical circuit analysis techniques, from the model in Figure 1 it can be easily shown [79]:

$$f_r = \frac{1}{2\pi\sqrt{L_r(C_r + C_s)}}, \tag{1}$$

$$Q_u = 2\pi f_r \frac{C_r + C_s}{G_r + G_s}, \tag{2}$$

thus, showing how the response of the resonator will be affected by the properties of the sample. Considering that $G_s$ and $C_s$ will depend on the complex dielectric permittivity of the sample (defined as $\varepsilon^*_{r,s} = \varepsilon_{r,s}' - j\varepsilon_{r,s}''$), which is linked to its glucose content, these devices therefore seem useful for glucose sensing. It is worthy to note that the dielectric properties of the unloaded planar resonators exhibit very low losses compared to the high dielectric losses of aqueous or biological glucose samples, and that this is a factor in favor of the sensitivity.

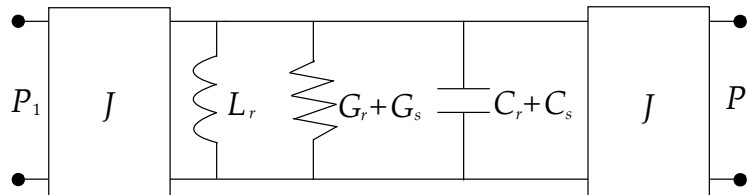

**Figure 1.** Parallel *RLC* lumped-element model of a microwave planar resonator loaded with a dielectric sample. The subscripts "r" and "s" refer to the contribution of the unloaded resonator and the sample to the respective element.

Under this approach, the first attempts for glucose level measurement with resonant methods started almost fifteen years ago [80,81]. These works and the above-mentioned interest in this technology motivated the research community to explore all sorts of ideas. Spiral-shaped resonators were investigated with aqueous glucose solutions [82,83] and biological phantoms [84]. Patch resonators [85], microstrip line-based resonators [86,87] or closed-loop rings [88] were also studied for glucose sensing. Notwithstanding these approaches, the most studied planar resonant solution has been the split-ring resonator (SRR)—also referred to as Open-Loop Resonator (OLR)—with a wide variety of configurations (e.g., [40,89–95]). Sensors based on SRR have been used for measurement of aqueous solutions [94,96,97], biological solutions [31,98] and measurements in human volunteers [39,99–101].

### 2.2. Proposal of Classification

Among the different possibilities for classification, in a wide application context, the working principle was proposed as the most convenient criterion [48], especially to ease the comparison. This classification yields five kinds of sensors: frequency-variation sensors (e.g., [94,101,102]), phase-variation sensors (e.g., [88,103]), frequency-splitting sensors (e.g., [104,105]), coupling-modulation sensors (e.g., [106–108]) and differential-mode sensors (e.g., [90,109,110]). This classification scheme, however, while interesting for a broad application view, might not be the most suitable one for the specific case of glucose sensing. Differentiating the sensors according to the operating principle, while coherent, might mask interesting complementary information that can be retrieved from different sensing parameters, which can be of interest for addressing sensitivity and, especially, selectivity issues.

In this sense, the focus seems to be put on the sensing parameter, i.e., the computed parameter from the sensor response which is eventually associated with the retrieved glucose concentration. The selected sensing parameter remarkably determines the sensing approach, even when remaining in the same sensing technique or working principle. In addition, several sensing parameters may be extracted from one single measurement. Accuracy enhancements by data fusion of several extracted and processed parameters from the same measured signal have been reported for other techniques [111]. Taking a brief look at the literature of microwave planar resonant glucose sensors, it is easy to see how several sensing parameters are considered in different references, such as $f_r$, $Q_u$, insertion/return loss or phase. However, the criteria for selecting them are not evident, and a consensus on which one (or ones) to use for each case does not seem to appear. All these reasons led us to propose a sensing parameter-based classification for the specific context of microwave planar resonant glucose sensors. The proposed sensor categorization scheme is summarized in Figure 2.

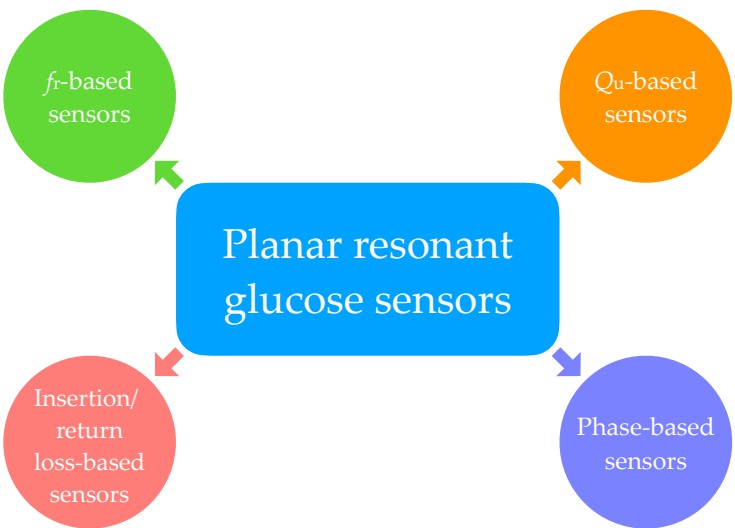

**Figure 2.** Proposed sensing parameter-based classification scheme.

The proposed classification allows us to make direct comparison between the sensors found in each category, since they all rely on the same sensing parameter, i.e., all the measurements are made in the same units, relying on the same physical processes. For example, in a broad application context aimed at dielectric permittivity measurements, for $f_r$-based sensors a normalized sensitivity relative to a percentage change in the permittivity of MUT was proposed in [102,112] as

$$S_{f_r} = \frac{\Delta f_r}{f_{r0} \cdot \Delta \varepsilon_r'} \times 100 \ [\%], \tag{3}$$

where $f_{r0}$ is the operating resonant frequency, $\Delta f_r = f_{r1} - f_{r0}$ is the measured resonant frequency shift ($f_{r1}$ is the absolute measured resonant frequency for a certain MUT) and $\Delta \varepsilon_r'$ is the dielectric permittivity change in the MUT with respect to the minimum measurable or considered permittivity (for general permittivity sensors it is usually considered $\Delta \varepsilon_r' = \varepsilon_{r,MUT}' - 1$).

Nevertheless, the main disadvantage of this classification criterion is the difficulty in making comparisons between different categories, where a change in the units is involved. Since the sensitivities found in the literature are heterogeneous and they depend on the sensing parameter, a common generalized metric seems desirable, which has not been hitherto proposed. As it may be of interest to analyze, compare and select sensing approaches with different sensing parameters, we propose an extension of the sensitivity concept introduced in (3) aimed to provide a generalized relative sensitivity which can be applied to different sensing parameters and to different application scenarios. For the sake of analysis, comparison and decision making, we suggest computing the sensitivity as the percentage change of the sensing parameter relative to a percentage change in the parameter under tracking. Consequently, we propose a Relative Sensitivity (*RS*) as

$$RS = \frac{\Delta SP}{SP_0 \cdot \Delta a} \times 100 \ [\%/\%], \tag{4}$$

where $a$ is the parameter to be measured. $\Delta a$ is the relative change in $a$ with respect to the lowest $a$ involved in the operating range of the sensor, expressed as a percentage (%). $SP_0$ is the operating value of the Sensing Parameter (*SP*)—i.e., the *SP* value obtained for the lowest $a$—and $\Delta SP = SP_1 - SP_0$ is the measured shift of the *SP* ($SP_1$ is the absolute measured value of the *SP* for a certain $a$). Introducing *RS* is relevant as it may enable the fair comparison of various sensors monitoring the same parameter using neither the same sensing parameter nor the same operating range for the tracked parameter. All other parameters being equivalent, higher *RS* means more sensitivity.

In our case, we compute the Relative Sensitivity to Glucose *RS(G)*, where $a = \rho_g$ is the glucose concentration expressed in wt% and *SP* is the sensing parameter to be related to the glucose concentration, which will be discussed throughout the paper. Therefore, according to the classification scheme in Figure 2, the next sections will review the most significant works on $f_r$-based, insertion/return loss-based, $Q_u$-based and phase-based planar resonant glucose sensors while applying the *RS(G)* concept. It is important to note that the *RS(G)* cannot be calculated for the specific case of phase-based sensors because of the difficulty in defining the $SP_0$ value. For example, a candidate parameter for this value could be the unwrapped phase of the sensor (absolute phase shift between the input and output of the device). However, it is not available in the references, and even if it was, it is still unclear if this would be a relevant calculation.

### 3. Resonant Frequency-Based Sensors

The sensors based upon resonant frequency as the sensing parameter to be related to the glucose content rely on the dependence of the resonant frequency on the relative effective permittivity ($\varepsilon_{r,eff}$). For example, for an open-loop half-wavelength transmission line resonator [79]:

$$f_r = \frac{c}{2l\sqrt{\varepsilon_{r,eff}}}, \tag{5}$$

where *c* is the speed of the light in free space and *l* is the length of the resonant line. This effective permittivity is indeed affected by the relative permittivity of the substrate and that of the media upon the sensor, where the glucose-containing sample is to be placed. Since the glucose concentration modifies the relative permittivity of the sample, the resonant frequency is expected to be affected by its changes. This can be more easily seen in (1) for the lumped-element model of a simple resonator loaded with a glucose-containing sample, where the influence on the sample is modeled by $C_s$. However, the relationship between the different relative permittivities involved is not trivial and it depends on the configuration. This aspect, in addition to the different possibilities for maximizing the sensitivity of the resonant frequency to variations in $\varepsilon_{r,eff}$, calls for research on $f_r$-based sensors. This section will review the most relevant planar resonant $f_r$-based glucose-sensing approaches of the last years.

As a general rule, some premises have been proposed to optimize the performances of these sensors. Firstly, as the quality factor is inversely related to the resonance bandwidth, a high $Q_u$ is convenient to obtain a high resolution in the measured $f_r$. It should be noted that this is a requirement related to the measurement process, and it does not imply a high sensitivity of $f_r$ to glucose concentration, since there is not a theoretical relationship between $Q_u$ and the $f_r$ sensitivity. Other convenient design aspects consist of reducing as much as possible the coupling capacitance between the I/O lines and the resonant elements, and using low-loss substrates [48]. Finally, highly capacitive configurations have been proved desirable for glucose level measurements [30,113].

Under these assumptions, the research community has made a number of attempts at and proposals for $f_r$-based glucose sensors. The most common one is the use of SRR with different configurations, which was recently studied in [114]. Single SRR configurations have been proposed with an extended capacitive gap [115], with an enzyme coating [91,96], with a microfluidic configuration measuring physiological glucose concentrations in water [116], with a CSRR-loaded square patch configuration [22,110,117–120], with a defected ground design for measuring physiological concentrations in blood plasma samples [98] or with an RF tag configuration able to measure physiological concentrations in phantom interstitial fluid [121] (Figure 3a). A double SRR on a gold-on-glass substrate was proposed in [122], able to measure physiological concentrations of glucose in water with a differential configuration. A structure based on open SRR was tested for microfluidic measurement [123]. More complex approaches combining a larger number of SRR have reported interesting sensitivity raises [41]. In order to increase the line capacitance of the sensor, interdigital capacitor gaps have been considered in different configurations

for water–glucose measurements [30,124,125]. The use of complementary electric *LC* resonators has also been reported for microfluidic applications with aqueous and biological samples [108,126] (see Figure 3b). Resonant transmission lines in microstrip [127] or coplanar [128,129] technology have also been proposed, as well as other resonant approaches such as epsilon negative unit-cell [130], planar whispering gallery mode [131] substrate-integrated waveguide [119] or corona shape [132].

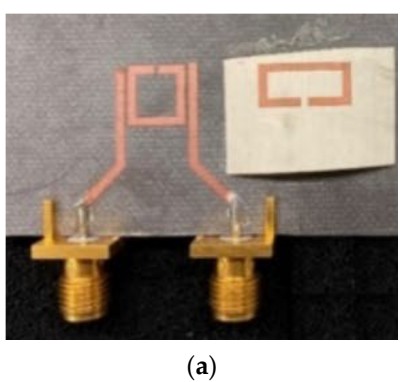
(**a**)
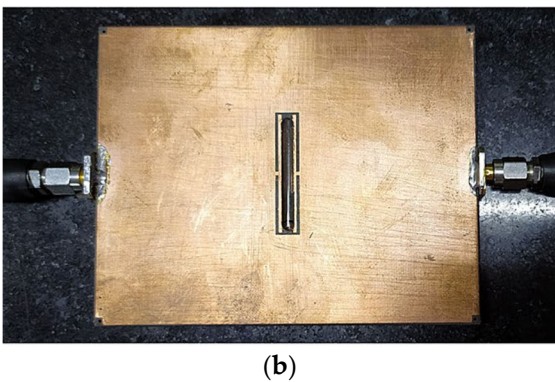
(**b**)

**Figure 3.** Examples of $f_r$-based sensors: (**a**) Single SRR sensor with RF tag sensor concept, reprinted from [121]; (**b**) Complementary *LC* resonator sensor integrating a microfluidic tube, reprinted from [126].

The compilation of these approaches along with their significant data and sensitivities is shown in Table 1, ordered by operating resonant frequency and type of sample (aqueous solution or blood). The sensitivities in KHz per mg/dL are given only for the cases in which measurements of physiological concentrations were shown. Data for some references had to be taken from the published plots and must be consequently considered as approximate. The *RS(G)* value was computed from the available data for all the references in the table, considering the lowest $\rho_g$ involved in each study for the computation of $\Delta\rho_g$. The *RS(G)* column will later enable further analysis and comparisons between various sensing parameter-based sensors.

**Table 1.** Resonant frequency-based sensors. AG = aqueous glucose solutions; (C) SRR = (complementary) split-ring resonator; (C) ELC = (complementary) electric *LC* resonator; ENG = epsilon negative; IDC = interdigital capacitor; MLIN = microstrip line; MF = microfluidic; SIW = substrate-integrated waveguide; TX = transmission; WGM = whispering gallery mode.

| Ref. | Sensor Type | Sample (Volume) | $\rho_g$ Range (wt%) | Bare $f_r$ (GHz) | Operating $f_r$ (GHz) | $S_{fr}$ (KHz per mg/dL) | RS(G) (%/%) | Remarks |
|------|------------|-----------------|---------------------|------------------|----------------------|--------------------------|-------------|---------|
| [124] | IDC-based resonator | AG (2 μL) | 0–8 | 2.46 | 0.64 | — | 0.556 | Biodegradable flexible substrate |
| [108] | Single CELC | AG (MF) | 0–10 | 1.40 | 1.16 | — | 0.408 | MF channel |
| [125] | IDC SRR | AG (MF) | 0–5 | 4.18 | 1.32 | — | 0.622 | MF channel |
| [122] | Differential double SRR | AG (MF) | 0.05–0.3 | 1.74 | 1.44 | 8.9 | 0.510 | Gold-on-glass substrate, MF |
| [91] | Single SRR | AG (90 μL) | 0–50 | 1.83 | 1.49 | — | 0.072 | Enzyme-coated |
| [96] | Single SRR | AG (90 μL) | 0–40 | 1.82 | 1.59 | — | 0.099 | Enzyme-coated |
| [41] | Triple CSRR | AG (1.2 mL) | 0.07–0.11 | 2.30 | 1.60 | 67.0 | 2.188 | MF channel |

**Table 1.** *Cont.*

| Ref. | Sensor Type | Sample (Volume) | $\rho_g$ Range (wt%) | Bare $f_r$ (GHz) | Operating $f_r$ (GHz) | $S_{fr}$ (KHz per mg/dL) | RS(G) (%/%) | Remarks |
|------|-------------|-----------------|---------------------|------------------|----------------------|-------------------------|-------------|---------|
| [126] | Single CELC | AG (MF) | 0.1–0.5 | 1.71 | 1.64 | 18.5 | 1.128 | Reusable, MF channel |
| [130] | ENG unit-cell resonator | AG (2 μL) | 0–10 | 2.09 | 1.91 | — | 0.895 | SRR- and horn-shaped elements |
| [127] | MLIN res. | AG (—) | 0–0.3 | — | 2.00 | 2.5 | 0.125 | — |
| [115] | Modified SRR | AG (MF) | 0–6 | — | 2.00 | — | 0.011 | Capacitive gap |
| [116] | Single CSRR | AG (MF) | 0–0.5; 0–8 | — | 2.48 | 5.0 | 0.202 | MF channel |
| [128] | Coplanar TX line | AG (50 μL) | 0–70 | 5.81 | 2.88 | — | 0.109 | Wireless system |
| [131] | Plasmonic, planar WGM | AG (MF) | 0–20 | 4.15 | 3.39 | — | 0.228 | MF channel |
| [30] | IDC resonator | AG (125 μL) | 0–1 | 4.80 | 3.43 | 14.0 | 0.408 | Pressure correction |
| [129] | Coplanar IDC | AG (15 μL) | 0–100 | 4.8 | 3.9 | — | 0.060 | Non-reciprocal |
| [119] | Circular SIW | AG (2.5 μL) | 0–30 | 4.40 | 4.33 | — | 0.089 | — |
| [117] | CSRR patch | AG (2.3 μL) | 0.3–0.7 | 5.00 | 4.39 | 6.8 | 0.155 | — |
| [132] | Corona res. | AG (—) | 0.1–0.5 | 6.25 | 7.01 | 7.25 | 0.103 | — |
| [123] | Triple open SRR | AG (MF) | 0–40 | 6.50 | 5.40 | — | 0.035 | MF channel |
| [126] | Single CELC | Goat blood (MF) | 0.1–0.5 | 1.71 | 1.64 | 56.0 | 3.415 | Reusable, MF channel |
| [121] | Tag single SRR | Mimicked ISF (200 μL) | 0–0.5 | 4.35 | 3.76 | 2.11 | 0.056 | RF tag sensor |
| [132] | Corona res. | Blood (—) | 0.1–0.5 | 6.25 | 7.01 | 3.50 | 0.050 | — |
| [98] | Single SRR | Blood plasma (—) | 0.09–0.15 | 8.33 | 8.32 | 123.08 | 1.479 | Defected ground |

## 4. Insertion/Return Loss-Based Sensors

The sensors tracking variations in the insertion or return losses are mainly (but not exclusively) based on the variations in the dielectric losses in the sample due to changes in the glucose concentration. Indeed, these losses, represented as $G_s$ in the model of Figure 1, have an impact on the conductance associated with the MUT, thus affecting the finally measured insertion/return losses. This effect is mostly associated with variations in $\varepsilon_{r,s}''$ linked to changes in the concentration of electrolytes in the sample [121]. These variations in the conductance of the MUT provoke variations in the amplitude of the measured response [133]. In terms of the scattering parameters, considering a simple two-port microstrip resonator, the magnitude return loss can be defined as [134]

$$\mathrm{mag}(S_{11}) = 20\log\left|\frac{R_{in} - Z_0}{R_{in} + Z_0}\right| = 20\log\left|\frac{1 - \frac{G_{in}}{Y_0}}{1 + \frac{G_{in}}{Y_0}}\right|, \tag{6}$$

where $Z_0$ and $Y_0$ are the effective impedance and admittance of the microstrip line with no sample, and $R_{in}$ and $G_{in}$ are the input resistance and conductance seen from port 1. For the model in Figure 1, the input admittance is:

$$Y_{in} = G_{in} + jB_{in} = G_r + G_s + j\left(\omega(C_r + C_s) - \frac{1}{\omega L_r}\right), \tag{7}$$

where $\omega$ is the angular frequency and $B_{\text{in}}$ is the input susceptance. The input conductance in (6) is therefore $G_{\text{in}} = G_{\text{r}} + G_{\text{s}}$.

A similar analysis can be applied for the insertion loss. The influence of the sample in the measured insertion or return loss is therefore evident, although it depends on the configuration. Some works have provided accurate descriptions of the return/insertion loss in terms of the lumped elements associated with the MUT for different configurations, such as a magnetically coupled SRR [135] or interdigital capacitor-based SRR [125]. The relationship between the targeted parameter and the sensing parameter is not so evident in this approach, and many attempts can be found in the literature trying to maximize the sensitivity by means of diverse configurations. The main drawback of these sensors is the sensitivity to noise effects and load mismatches, which can pose a considerable challenge when highly accurate insertion/return loss measurements are required, especially for commercial, portable applications. Notwithstanding, these sensors can require notably narrow band interrogation signals for practical measurements, thereby easing the associated driving and data acquisition electronics.

This has been a common sensing approach leading to a sizable number of publications. Some of the $f_{\text{r}}$-based attempts have also been considered with these sensing parameters, providing in some cases for interesting dual measurement capabilities [121,127,129,130,132] (Figure 4a). Again, the SRR is present in many of the proposed designs. Single SRR were considered with different designs for water–glucose solutions [94,136,137], aqueous microfluidic [135,138] and blood plasma [31] measurements. Additionally, the influence of ambient temperature on glucose concentration retrieval and a correction strategy were studied in [135], and additional environmental effect elimination was provided in [118] with a dual SRR approach. Two SRR for differential mode measurements, based on the splitter/combiner microstrip structure [104], were proposed in [92] for aqueous microfluidic measurement of physiological concentrations. An approach based on inter-resonator coupling of two mutually coupled SRR was studied for water–glucose solutions measurement [105]. Proposals combining larger amounts of SRR have shown promising results able to address physiological concentrations in aqueous solutions [120]. Considering other configurations, a coplanar electric *LC* resonator was proposed for glucose concentration measurement in watery and PBS solutions [139]. A circular SIW structure was used for measuring water–glucose solutions [119]. Physiological glucose concentrations in aqueous solutions were successfully tracked with a T-shaped microstrip line [140], with a quarter-wavelength stub connected to an interdigital capacitor structure in a microfluidic setup [141] (Figure 4b), and with a microstrip line resonator [86].

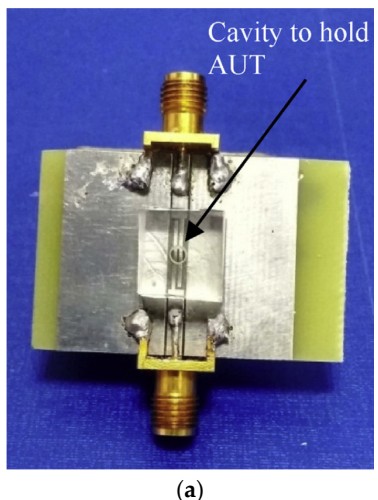

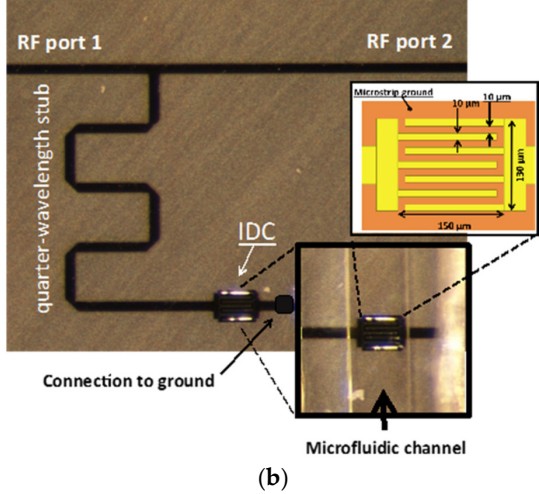

(**a**)  (**b**)

**Figure 4.** Examples of insertion/return loss-based sensors: (**a**) Coplanar interdigital capacitor sensor, reprinted from [129] with permission from Elsevier; (**b**) Quarter-wavelength stub connected to an interdigital capacitor structure for microfluidic measurements, reprinted from [141].

The review for this approach is summarized in Table 2 with the references ordered by operating resonant frequency and type of sample. The sensitivities in dB per mg/dL are given only for the cases in which measurements of physiological concentrations were shown. Data for some references had to be taken from the published plots and must be consequently considered as approximate. As before, the *RS(G)* value was computed from the available data for all the references in the table, considering the lowest $\rho_g$ for each one as reference.

**Table 2.** Insertion/return loss-based sensors. AG = aqueous glucose solutions; (C)ELC = (complementary) electric *LC* resonator; IDC = interdigital capacitor; MLIN = microstrip line; SRR = split-ring resonator; TX = transmission.

| Ref. | Sensor Type | Sample (Volume) | $\rho_g$ Range (wt%) | Op. $f$ (GHz) | Bare $S_{11}/S_{21}$ (dB) | Operating $S_{11}/S_{21}$ (dB) | $S_{S11}/S_{S21}$ (dB Per mg/dL) | $RS(G)$ (%/%) | Remarks |
|---|---|---|---|---|---|---|---|---|---|
| [92] | Differential double SRR | AG (MF) | 0–10 | 0.75 | — | −12.75 | $0.55 \times 10^{-3}$ | 4.314 | MF channel, differential |
| [120] | Four SRR structure | AG (7.5 mL) | 0–0.3 | 1.80 | −12.65 | −10.75 | $0.42 \times 10^{-3}$ | 3.923 | Portable solution |
| [136] | Single SRR | AG (20 µL) | 0–20 | 1.96 | −17.20 | −16.40 | — | 0.264 | — |
| [127] | MLIN res. | AG (—) | 0–0.3 | 2.00 | — | −11.17 | $0.12 \times 10^{-3}$ | 1.044 | — |
| [130] | ENG unit-cell resonator | AG (2 µL) | 0–10 | 2.09 | −21.00 | −19.60 | — | 0.506 | SRR- and horn-shaped elements |
| [137] | Single SRR | AG (20 µL) | 0–100 | 2.41 | −12.80 | −13.20 | — | 0.386 | — |
| [135] | Single CSRR | AG (MF) | 0–0.4 | 2.42 | −28.00 | −22.00 | $0.08 \times 10^{-3}$ | 0.341 | Temp. corrected, MF channel |
| [118] | Double CSRR | AG (MF) | 0–0.4 | 2.42 | −28.00 | −16.80 | $0.08 \times 10^{-3}$ | 0.446 | Temp. and humidity corr., MF |
| [139] | Coplanar ELC | AG (20 µL) | 4–20 | 3.41 | −19.50 | −4.80 | — | 0.651 | — |
| [129] | Coplanar IDC | AG (15 µL) | 0–100 | 3.90 | −27.50 | −18.00 | — | 0.850 | Non-reciprocal |
| [105] | Two coupled SRR | AG (5 µL) | 0–10 | 4.23 | −7.21 | −10.48 | — | 3.244 | Inter-resonators coupling |
| [119] | Circular SIW | AG (2.5 µL) | 0–30 | 4.33 | −4.63 | −9.16 | — | 1.380 | — |
| [86] | MLIN res. | AG (7.5 mL) | 0.08–5 | 4.88 | — | −22.1 | $0.52 \times 10^{-3}$ | 2.344 | — |
| [94] | Single SRR | AG (25 µL) | 0–10 | 5.16 | −14.27 | −25.50 | — | 0.329 | — |
| [140] | T-shaped line | AG (0.6 mL) | 0–0.6 | 6.00 | — | −15.00 | $0.54 \times 10^{-3}$ | 3.600 | Finger shaped |
| [141] | λ/4 stub with IDC structure | AG (MF) | 0–8 | 7.50 | −17.80 | −9.00 | $0.72 \times 10^{-3}$ | 4.889 | MF channel |
| [138] | Coplanar single SRR | AG (MF) | 0–1.2 | 18.63 | −24.85 | −19.84 | $0.23 \times 10^{-3}$ | 1.159 | Chromium-gold layer, MF |

**Table 2.** *Cont.*

| Ref. | Sensor Type | Sample (Volume) | $\rho_g$ Range (wt%) | Op. $f$ (GHz) | Bare $S_{11}/S_{21}$ (dB) | Operating $S_{11}/S_{21}$ (dB) | $S_{S11}/S_{S21}$ (dB Per mg/dL) | RS(G) (%/%) | Remarks |
|------|-------------|-----------------|----------|------|------|------|------|------|---------|
| [139] | Coplanar ELC | PBS (20 µL) | 4–20 | 3.41 | −19.5 | −3.90 | — | 0.321 | — |
| [121] | Tag single SRR | Mimicked ISF (200 µL) | 0–0.5 | 3.76 | −41.00 | −56.00 | $0.83 \times 10^{-3}$ | 1.486 | RF tag sensor |
| [31] | Single SRR | Blood plasma (25 µL) | 0–10 | 5.17 | −14.27 | −25.08 | — | 0.163 | Multicomponent solutions study |
| [132] | Corona res. | Blood (—) | 0.1–0.5 | 7.01 | — | −27.50 | $0.57 \times 10^{-3}$ | 2.045 | — |

## 5. Quality Factor-Based Sensors

The use of the quality factor in resonant sensors for measurements of variations in the dielectric properties of the MUT was proposed and studied some years ago, especially for lossy solutions [81,107,142]. However, it was not until recent years that this approach started to be exploited for glucose concentration measurement with planar sensors [31,94,105,143]. Indeed, the measured loaded quality factor ($Q_L$) is defined as

$$Q_L = \frac{f_r}{\Delta f_{3dB}}, \tag{8}$$

where $f_r$ is the resonant frequency and $\Delta f_{3dB}$ is the measured bandwidth at 3 dB fall from the amplitude maximum at the resonant frequency. The influence of the loading of the ports due to the feed and measurement equipment (usually a Vector Network Analyzer) is involved in the computation of $Q_L$. To avoid it, for a two-port planar resonator with a low enough I/O coupling, the unloaded quality factor ($Q_u$) can be computed as:

$$Q_u = \frac{Q_L}{1 - 10^{\frac{S_{21max}}{20}}}, \tag{9}$$

where $S_{21max}$ is the maximum amplitude (in dB) of the transmission parameter, given at the resonant frequency. Therefore, Equations (8) and (9) show how this approach seeks to benefit from the information included both in $f_r$ and $S_{21max}$ (two prior sections) and it also introduces a new source of information, $\Delta f_{3dB}$, which can also be useful for tracking effects specifically associated with changes in the glucose concentration [90].

Going back to the illustrative lumped-element example in Figure 1, for a homogeneous dielectric sample at a sufficiently high frequency to neglect DC conductivity, its loss tangent ($\tan \delta_s = \varepsilon_{r,s}'' / \varepsilon_{r,s}'$) at the resonant frequency can be expressed in terms of its lumped elements:

$$\tan \delta_s = \frac{G_s}{2\pi f_r C_s}. \tag{10}$$

Hence, the definition of $Q_u$ for the model in Figure 1, given in (2), can be rearranged and expressed as a function of $\tan \delta_s$:

$$Q_u = \frac{\frac{C_r}{C_s} + 1}{\tan \delta_s \left(\frac{G_r}{G_s} + 1\right)}, \tag{11}$$

thus showing the relationship of this parameter with the sample loss tangent. This analysis shows coherence with the conclusions reached in [144] for a microstrip patch resonator.

Equation (11) therefore shows how $Q_u$-based sensors are related to variations both in real and imaginary parts of the permittivity of the sample without ranking or giving

relative qualification between them. Indeed, their dependence on the loss tangent of the MUT seems interesting for tracking glucose concentration changes [77]. Actually, recent studies of the dielectric signature of glucose solutions showed the convenience of tracking parameters influenced by the dielectric losses of the MUT at microwave frequencies up to roughly 10 GHz [71]. In fact, the loss factor has also been proposed for measuring other biomedical water-based processes at these frequencies [145].

As a matter of fact, the importance of properly tracking loss tangent variations in microwave sensors aimed to glucose concentration retrieval was pointed out in [146]. With this approach, non-planar techniques have been proposed with this $Q_u$-based configuration for aqueous glucose characterization, such as whispering-gallery-mode resonators [147,148], although requiring bulky setups. As for planar devices for glucose concentration measurement with this approach, the use of SRR has been commonly considered. An SRR loaded with a coplanar waveguide line was used in [143] for measurements in aqueous solutions. An asymmetric SRR for aqueous glucose solutions was proposed in [149]. The use of a single SRR with a dielectric sample holder was studied for water–glucose [94] (Figure 5a) and blood plasma–glucose [31] solutions, pointing the challenges when multicomponent solutions are involved. A significant sensitivity raise was reported for a novel technique based on two mutually coupled SRR [105] (Figure 5b). Another work proposed an open-loop line resonator for water–glucose solutions [81]. A more compact solution was shown with a coplanar quarter-wavelength resonator [150], which included a study regarding the effect of the temperature upon the finally obtained sensitivity.

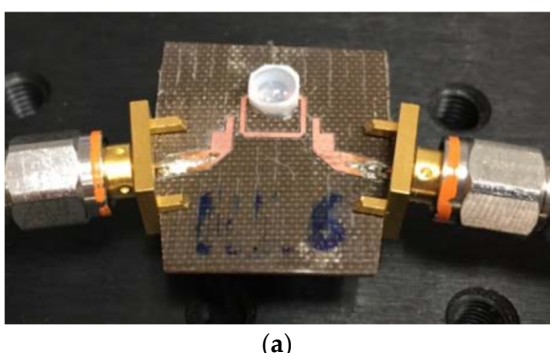 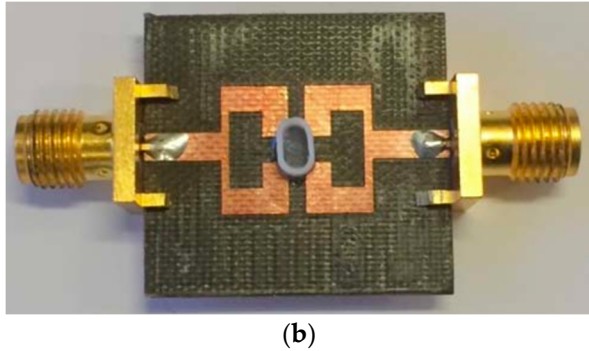

(**a**) (**b**)

**Figure 5.** Examples of $Q_u$-based sensors: (**a**) Single SRR sensor with dielectric sample holder. © (2019) IEEE. Reprinted, with permission, from [94]. (**b**) Pair of mutually coupled SRR exploiting inter-resonators coupling. © (2019) IEEE. Reprinted, with permission, from [105].

Table 3 shows a summary for the review of this sensing approach, with all the works in order considering the operating resonant frequency and type of sample. Although no references have hitherto shown measurements of physiological concentrations using this approach, the corresponding column is kept in the table (with no data) to maintain consistency and ease the global comparison. Data for some references had to be taken from the published plots and must be consequently considered as approximate. Following the general analysis, the *RS(G)* value was computed from the available data for all the references in the table, considering the lowest $\rho_g$ for each one as reference. The reference [105] was separated since it does not involve a strictly $Q_u$-based design technique, but an inter-resonator coupling one. However, the computation of a virtual quality factor was proposed and used for glucose concentration tracking with interesting results, and it also deserves careful attention.

**Table 3.** Quality factor-based sensors. AG = aqueous glucose solutions; WG = waveguide.

| Ref. | Sensor Type | Sample (Volume) | $\rho_g$ Range (wt%) | Op. $f$ (GHz) | Bare $Q_u$ | Operating $Q_u$ | $S_{Qu}$ (Per mg/dL) | RS(G) (%/%) | Remarks |
|------|-------------|-----------------|----------------------|---------------|------------|-----------------|----------------------|-------------|---------|
| [81] | Open-loop line resonator | AG (5 μL) | 0–4 | 1.11 | — | 187.14 | — | 0.186 | — |
| [143] | Single SRR | AG (200 μL) | 0–20 | 2.45 | — | 60.00 | — | 0.948 | Coplanar WG |
| [149] | Asymmetric single SRR | AG (—) | 0–5 | 4.50 | 118.78 | 116.18 | — | 1.511 | — |
| [150] | Coplanar λ/4 resonator | AG (5 μL) | 0–10 | 4.79 | 96.80 | 85.46 | — | 0.901 | Temperature effect analysis |
| [94] | Single SRR | AG (25 μL) | 0–10 | 5.16 | 112.31 | 60.65 | — | 0.978 | — |
| [94] | Single SRR | AG (5 μL) | 0–10 | 7.16 | 109.27 | 72.68 | — | 0.584 | — |
| [31] | Single SRR | Blood plasma (25 μL) | 0–10 | 5.17 | 112.31 | 58.16 | — | 0.829 | Multicomponent solutions study |
| [31] | Single SRR | Blood plasma (5 μL) | 0–10 | 7.17 | 109.27 | 64.99 | — | 0.571 | Multicomponent solutions study |
| [105] | Two coupled SRR | AG (5 μL) | 0–10 | 4.23 | 89.85 | 47.63 | — | 4.115 | Inter-resonators coupling |

## 6. Phase-Based Sensors

Planar sensors based upon the measured phase-shift for tracking the glucose concentration have also been reported in the literature, although sparingly. These sensors have been proposed for multiple applications requiring the measurement of the dielectric properties of the MUT [151]. These sensors show the advantage of requiring very narrow band interrogation signals (even a single tone in some cases), and the associated driving electronics become simpler and more cost-effective. Nevertheless, the data acquisition systems for phase-based measurements can become considerably complex in comparison with other measurement approaches. In this sense, solutions based on the measurement of the magnitude of the transmission coefficient associated with a phase difference sensing have been recently proposed [152], making the phase-based approach interesting from the practical realization point of view. All these reasons have led some researchers to consider the application of this kind of sensor for glucose level identification.

The working principle of these sensors is similar to the one of the insertion/return loss-based sensors, but with the focus put on the phase of the transmission or reflection coefficient. Effectively, the reflection coefficient of the lumped-element model in Figure 1, without considering the admittance inverters for simplicity, can be expressed as

$$S_{11} = \frac{Z_{in} - Z_0}{Z_{in} + Z_0} = \frac{1 - \frac{Y_{in}}{Y_0}}{1 + \frac{Y_{in}}{Y_0}}, \tag{12}$$

where $Z_{in}$ is the input impedance seen from port 1. Then, considering the definition of $Y_{in} = G_{in} + jB_{in}$, the phase of the reflection coefficient can be computed as

$$\phi_{S_{11}} = \arctan\left(-\frac{2B_{in}Y_0}{Y_0^2 - G_{in}^2 - B_{in}^2}\right). \tag{13}$$

Finally, for the model considered, applying (7) leads to

$$\phi_{S_{11}} = \arctan\left(\frac{\frac{2Y_0}{\omega L_r}(1 - \omega^2 L_r(C_r + C_s))}{Y_0^2 - (G_r + G_s)^2 - \frac{1}{\omega^2 L_r^2}(\omega^2 L_r(C_r + C_s) - 1)^2}\right), \tag{14}$$

where the influence of the glucose-containing sample in the finally measured phase can be seen through $C_s$ and $G_s$ elements.

A similar analysis can be applied to a transmission line, in which the electrical length of a section of microstrip line of physical length $l$ is defined as $\theta = kl$, $k = 2\pi f_r/v_{p0}$ being the phase constant, and $v_{p0}$ the phase velocity at the resonant frequency. This phase velocity is given by

$$v_{p0} = \frac{c}{\sqrt{\varepsilon_{r,\text{eff}}}}. \tag{15}$$

Since the phase velocity depends on the effective permittivity, a change in $\varepsilon_{r,\text{eff}}$ results in a change in the phase of the reflection coefficient. This, thus, leads to a similar working principle to that for the $f_r$-based sensors presented in Section 3, but with the measured phase as a sensing parameter.

The use of phase-based sensors with SRR-based approaches has been reported in the literature for dielectric properties measurement [103], although this has not been widely applied for glucose sensing. A microfluidic attempt considering a triple SRR design was proposed in [123] for aqueous glucose samples, combined with $f_r$-based measurements. In [86] a microstrip resonator was used for glucose sensing in aqueous solutions at physiological concentrations, combining phase and insertion loss measurements. An interesting design aimed at measurements in human fingers was presented in that work. A similar microstrip line-based approach was studied in [153] by means of full-wave simulations, hitherto lacking experimental validation. A microstrip line-based sensor was proposed in [154] for measuring aqueous glucose solutions at several frequencies using a pseudo-random noise driving system. A planar closed-loop ring resonator was shown to provide interesting sensitivity and the ability to address measurement of physiological concentrations in water–glucose solutions [88].

From the point of view of sensitivity normalization, phase-based sensors constitute a special case since the selection of the operating sensing parameter ($SP_0$) in (4) to compute the $RS(G)$ is not trivial. Indeed, taking as reference the measured phase for the lowest $\rho_g$ case does not make much sense, since it gives no information about the performance of the sensor, and could lead to confusing percentage sensitivities. In ref. [155], a figure of merit was proposed for phase-based dielectric sensors, expressed as the ration between the maximum sensitivity in ° and the size of the sensing region in terms of the squared guided wavelength. While consistent, this parameter seems not to provide the desired information about the sensor performance for the specific case of glucose sensing. An interesting value for $SP_0$ could be the measured phase difference for the lowest $\rho_g$ value, which is, in general, unfortunately neither given in the references, nor possible to compute from the provided data. Due to these reasons, the review for glucose planar phase-based sensors is summarized in Table 4, ordered by operating frequency, without any computation of $RS(G)$, providing the raw sensitivity in terms of ° per wt%. The sensitivities in ° per mg/dL are given only for the cases in which measurements of physiological concentrations were shown. The operating phase for the reflection or transmission parameter is given for the lowest $\rho_g$ case for informational purposes only.

**Table 4.** Phase-based sensors. AG = aqueous glucose solutions; MLIN = microstrip line; PRN = Pseudo random noise.

| Ref. | Sensor Type | Sample (Volume) | $\rho_g$ Range (wt%) | Op. $f$ (GHz) | Operating $\phi_{S11}/\phi_{S21}$ (°) | $S_{\phi S11}/S_{\phi S21}$ (° per mg/dL) | $S_{\phi S11}/S_{\phi S21}$ (° per wt%) | Remarks |
|------|-------------|-----------------|---------------------|---------------|-------------------------------|-------------------------------------|------------------------------------|---------|
| [88] | Closed-loop ring | AG (—) | 0–0.25 | 4.02 | −66.00 | $4.694 \times 10^{-3}$ | 4.694 | — |
| [86] | MLIN res. | AG (7.5 mL) | 0.08–5 | 4.88 | −111.43 | $1.510 \times 10^{-3}$ | 1.510 | — |
| [123] | Triple open SRR | AG (MF) | 0–40 | 5.40 | 134.38 | — | 7.813 | MF channel |
| [154] | MLIN-based sensor | AG (—) | 0–5 | 7.81 | — | $0.037 \times 10^{-3}$ | 0.037 | Portable, PRN-driven system |
| [153] | MLIN res. | AG (—) | 0–0.11 | 19.04 | −82.79 | $3.182 \times 10^{-3}$ | 3.182 | Only simulations |

## 7. Discussion

The use of planar microwave resonant sensors for dielectric properties characterization has been widely studied during the last decades. Their unique features in terms of ease of integration and cost-effectiveness, as well as frequency range of operation, have drawn the attention of a considerable part of the scientific community willing to develop glucose concentration sensors. The development of the topic has led to numerous publications during the last years, showing a wide variety of designs, contexts of use and results. At this point, a review of the most relevant ones seems convenient.

While relying on the same technological principles, different ways of extracting the information from the measurement have been proposed, basing on different sensing parameters. This is why, once the technology and working principle have been narrowed down, this review proposes a classification according to the sensing parameter. This, in fact, can be understood as a subcategorization within more general, larger classification schemes based on technological working principles [48], focused on the specific context of glucose sensors.

However, whereas sorting and displaying the approaches according to the sensing parameter can be useful for comparison and discussion within each category, the general, cross-type study becomes intricate in the absence of common performance indicators. To overcome this issue, a general relative sensitivity to glucose *RS(G)* has been defined and applied to all the analyzed references. In a general glucose measurement context, where not only physiological measurements are involved, the *RS(G)* proposes a standard tool that normalizes the obtained sensitivity regardless of the sensing parameter relative to an also normalized glucose content (1 wt%). It is worthwhile to note that the *RS(G)* implicitly implies a penalty for sensors operating at a relatively high value of their sensing parameter, as is usual for any normalizing tool. This can be interesting to account for the extra difficulties in real devices to achieve high resonant frequencies or quality factors, or measuring too low values for the transmission coefficient, for example. In this sense, although convenient for general analysis and comparison, careful study of each attempt in its specific context should always be considered.

The sensors measuring a shift in resonant frequency find a wide variety of applications, and their use in the glucose context has been considered in many cases. Their working principle is mainly based on the changes in the real part of the permittivity of the sample. The use of SRR-based configurations has been reported in different cases with good results, especially when two or three of them are considered in the design. Additionally, complimentary electric *LC* resonators have shown promising results both in aqueous and biological solutions. These sensors often require high-quality factors for proper functioning, especially for low concentration changes detection, since a sharp resonant peak (or notch) is needed to allow distinction of slight frequency shifts. Active methods can be applied to enhance this aspect and raise the final practical sensitivity. While simple and easy to design, these sensors might require extremely high frequency resolution for low concentrations and wide band interrogation signals for high concentrations detection, which can lead to complex and costly final systems, depending on the application.

Insertion/return loss-based sensors rely on similar principles but extract the information from amplitude changes, which are more related to the losses in the sample. Some dual approaches have been reported as able to relate the glucose content changes to both the measured resonant frequency and insertion/return loss, which are of interest from the point of view of robustness. Within this approach, in addition to SRR-based designs, interdigital capacitor elements have been also considered in some cases with promising results. These sensors can achieve high *RS(G)* values, often greater than those from $f_r$-based sensors, both for aqueous and biological solutions. Additionally, this sensing approach may require narrow interrogation signals (which could be even reduced to a single tone, depending on the case), thus easing the final implementation. However, this sensing parameter is especially sensitive to noise, electronic calibration and load mismatch effects, which can hinder the detection process when high sensitivities and resolutions are targeted.

The unloaded quality factor, which finds application in other measurement contexts, has also been proposed for glucose measurements, and some $Q_u$-based approaches have been studied in recent years. Indeed, according to (8) and (9), this attempt combines the parameters considered in the previous two ones in addition to the measured bandwidth (which was also proved to be convenient for glucose measurements [90]), searching for a more robust estimation of the glucose level. It is therefore interesting to study the individual influence of $f_r$, $\Delta f_{3dB}$ and $S_{21max}$ in the finally obtained $Q_u$. Let us consider, for simplicity, the maximum amplitude of the transmission parameter in linear scale: $S_{21max,\text{lin}} = 10^{S21max/20}$. Then, applying (8) into (9) and solving for the partial derivatives yields

$$\frac{\partial Q_u}{\partial f_r} = \frac{1}{\Delta f_{3dB}(1 - S_{21max,\text{lin}})}, \tag{16}$$

$$\frac{\partial Q_u}{\partial \Delta f_{3dB}} = \frac{-Q_u}{\Delta f_{3dB}}, \tag{17}$$

$$\frac{\partial Q_u}{\partial S_{21max,\text{lin}}} = \frac{Q_u}{(1 - S_{21max,\text{lin}})}. \tag{18}$$

This means that, for a sensor with a given value of $Q_u$, the variations in this parameter will be more sensitive to changes in: $\Delta f_{3dB}$ for narrower $\Delta f_{3dB}$ (17); $S_{21max,\text{lin}}$ for greater $S_{21max,\text{lin}}$ (18), i.e., greater amplitude levels; and $f_r$ for a proper combination of the prior cases (16). In other words, the designs relying on this sensing approach can benefit from the information contained in $f_r$, $\Delta f_{3dB}$ and $S_{21max}$, and the influence of each parameter can be adjusted from the design, thus adapting to different sensing scenarios.

This approach has also been applied with SRR-based designs for measurements with watery and biological solutions, although other proposals have been made considering open-loop line resonators and coplanar designs. The attained *RS(G)* values show good comparative results with the rest of approaches, and a considerable sensitivity raise was reported recently for coupled SRR techniques, as it has also been shown for some insertion loss-based attempts. Despite that, no measurements of physiological glucose concentrations have been hitherto reported with this approach. These sensors may require moderately narrow interrogation signals, especially for low $f_r$ variations, although it depends on each design and application. They are also expected to provide more robust measurements. The measurements with this approach are partially related to the dielectric losses in the sample. The main interest of this sensing parameter is that it allows us to take into account changes in both the real and imaginary parts of the permittivity of the sample. Conversely, this approach requires the use of proper RF detection techniques, which can become complex in some cases, and further postprocessing of the measured signal to obtain the desired information.

Regarding the phase-based sensors, a few attempts have been reported in the literature. The *RS(G)* concept seems not to be directly applicable to this sensing approach with the available data in the literature, and therefore solely the raw sensitivities have been included in this review. Only water–glucose solutions have been considered with these sensors, most of them addressing physiological concentrations. The use of microstrip line-based designs has been involved in some attempts, although closed-loop resonators and SRR-based designs have also proved good comparative sensitivities. These sensors can operate at a single frequency, thereby easing the development of the required driving system and thus reducing the associated expense, although sophisticated phase-detection stages are required (or other hybrid solutions [152]). Recently, the properties of meandered lines have been proved to enhance the sensitivity of this kind of sensor for dielectric characteristics retrieval [156], as well as capacitively loaded lines [157], concepts that have contrarily not been hitherto applied to glucose sensing. This sensing approach remains therefore open for further study and exploitation, and its development could lead to interesting complementary measurements in the future.

As a general consideration, it is important to note that both real and imaginary parts of the permittivity change depending on the frequency, and therefore the operating frequency must be chosen carefully, also considering the rest of practical constraints. The references considered in this review have been ordered by the operating frequency, but this parameter is not considered in the *RS(G)* calculation (only for $f_r$-based sensors). Therefore, conclusions drawn from *RS(G)* values for each specific sensor should be considered in a moderate way, since the frequency could have an impact on the sensitivity. That said, at the moment, considering the general reported *RS(G)* values, it seems that insertion/return loss-based sensors are more sensitive than $f_r$-based ones. Additionally, $Q_u$-based sensors report generally higher *SR(G)* values than $f_r$-based ones, often similar to those from insertion/return loss-based approaches. These data prove how important it is to study, analyze and properly select the sensing parameter from the design stages.

Additionally, it is interesting to analyze the information provided by each sensing parameter. In this sense, according to (5), $f_r$ is of interest for tracking considerable changes in $\varepsilon_{r,s}'$. The insertion/return losses, however, seem more interesting for detecting the variations in $\varepsilon_{r,s}''$, as shown in (7). Finally, according to (11) and (14), $Q_u$- and phase-based approaches seem interesting for measuring combinations of both variations. These conclusions, in fact, come from the theoretical analysis of the generic model, and they are not strictly linked to the nature of the sample. This discussion is therefore applicable to any sensing scenario considering planar resonant elements, whatever the MUT. Therefore, the sensing parameter should be accordingly chosen depending on the expected variations of the dielectric characteristics of the samples.

In general, it can be seen how microwave planar glucose sensors show the general trend of considering SRR in the designs, chiefly due to its high capacitive, permittivity-dependent characteristics. Other elements, such as interdigital capacitors or electric *LC* resonators, have also been proved to yield good results. During recent years the convenience of considering mutually coupled resonators has also raised some attention. Regardless of the design, these sensors are convenient for their simplicity, ease of integration and even often for their non-invasive capabilities. However, they are also notably affected by undesirable cross-sensitivities to certain environmental factors, such as temperature, pressure and humidity, as well as intrinsic factors of the measurement context such as variations in other components different to glucose. Due to these reasons, their real use would require accurate calibration and, most likely, redundant information acquisition. Therefore, the analysis of the possibilities to extract information from different sensing parameters and its implications seem of interest.

## 8. Conclusions

The analysis of the available literature on microwave planar resonant sensors for glucose concentration detection reveals intense scientific activity during the last decade, even more noticeable during the last few years. Whereas the fundamentals of these sensors are based on general dielectric properties measurement techniques with resonant methods, the unique features of the targeted input variable and final application make their contextualized, in-depth study worthwhile. This work provides an updated review of the most significant attempts published within this approach, sorted by the sensing parameter. To ease the comparison and the general understanding of the different proposals, a common relative sensitivity has been proposed and applied to most of the references involved, which enables even the comparison between sensors belonging to different categories.

The general discussion points to some elements, such as SRR, interdigital capacitors or electric *LC* resonant designs, that have been commonly used to attain interesting sensitivities. Indeed, the reported sensitivities show optimism in some cases, for all of the sensing parameters. These sensitivities can be even further enhanced by means of recently developed active circuitry or machine-learning techniques. The selectivity, however, remains a challenging issue still requiring further clarification and development. In this sense, the acquisition of redundant information could be of interest for future multisensor

systems aiming at more robust, more selective measurements. Considering the sensitivities reported and the integration capabilities, planar resonant sensors could play a noticeable role in such systems. Therefore, the selection of the proper approaches from the early design stages becomes essential, and thus the availability of clear comparison tools seems convenient. This, in addition to the interest in obtaining complementary information from the measurements, becomes the main motivation to provide a classification of the sensors according to sensing parameter whilst considering a general relative sensitivity to glucose.

**Funding:** The work by Carlos G. Juan was supported by SAD Région Bretagne, No-Needle Project.

**Institutional Review Board Statement:** Not applicable.

**Informed Consent Statement:** Not applicable.

**Data Availability Statement:** Not applicable.

**Conflicts of Interest:** The authors declare no conflict of interest.

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
