# Peer review of "Microwave Planar Resonant Solutions for Glucose Concentration Sensing: A Systematic Review"

_applsci, doi:10.3390/app11157018_

Round 1

Reviewer 1 Report

The authors provide an extensive review of the literature for glucose concentration sensing using planar microwave resonant sensors. One of the key highlights in this study is the “Relative Sensitivity to Glucose” parameter denoted RS(G) that the authors introduced in the manuscript. The reviewer believes that it is important to compare different techniques that could otherwise not be compared quantitatively. This has been achieved for frequency-based, insertion/return loss based and quality factor-based approaches. The review also looks at phase-based approaches which could not be classified with the newly introduced RS(G) parameter. I would recommend this work for publication if the following comments are addressed:

  • Most importantly, the authors should spend more time explaining the RS(G) parameter. For example, does a higher RS(G) mean better sensitivity? What are the limits of this parameter? It would be beneficial to show one or more example calculations using values reported in Tables 1, 2 or 3.
  • Equation 4 notation should be revisited, the reviewer believes there might be a mistake here unless RS has units of [1/%]. The term Δa[%] is unclear, please provide this term explicitly.
  • Lines 57-60: Industrial processes use sensors that can be classified as inline, atline, offline, etc. These terms could help explain this further.
  • Line 158: Section 2.2 title has a typo
  • Line 458: typo in “neither”

Reviewer 2 Report

The paper presents an up-to-date review of glucose solution sensing approaches after more than one decade of research and development. Especially, the paper discusses the sensing parameter and the key points of different sensing categories. The manuscript is well prepared, and the reviewer has mainly the following question. 

What kind of planar resonant sensors the authors would like to recommend for glucose solution measurements ? To the reviewer's knowledge, resonant method is often used to measure the dielectric properties of low loss material. Does the reviewer agree with this? Is there any limitation of the planar resonant microwave sensors? What frequency range should be considered for the glucose solution measurements and why ?
